# Auxin and Root Gravitropism: Addressing Basic Cellular Processes by Exploiting a Defined Growth Response

**DOI:** 10.3390/ijms22052749

**Published:** 2021-03-09

**Authors:** Nataliia Konstantinova, Barbara Korbei, Christian Luschnig

**Affiliations:** Department of Applied Genetics and Cell Biology, Institute of Molecular Plant Biology, University of Natural Resources and Life Sciences, Vienna (BOKU), Muthgasse 18, 1190 Wien, Austria; nataliia.konstantinova@boku.ac.at (N.K.); barbara.korbei@boku.ac.at (B.K.)

**Keywords:** auxin, polar auxin transport, root gravitropism, PIN-FORMED

## Abstract

Root architecture and growth are decisive for crop performance and yield, and thus a highly topical research field in plant sciences. The root system of the model plant *Arabidopsis thaliana* is the ideal system to obtain insights into fundamental key parameters and molecular players involved in underlying regulatory circuits of root growth, particularly in responses to environmental stimuli. Root gravitropism, directional growth along the gravity, in particular represents a highly sensitive readout, suitable to study adjustments in polar auxin transport and to identify molecular determinants involved. This review strives to summarize and give an overview into the function of PIN-FORMED auxin transport proteins, emphasizing on their sorting and polarity control. As there already is an abundance of information, the focus lies in integrating this wealth of information on mechanisms and pathways. This overview of a highly dynamic and complex field highlights recent developments in understanding the role of auxin in higher plants. Specifically, it exemplifies, how analysis of a single, defined growth response contributes to our understanding of basic cellular processes in general.

## 1. Background—At the Root of it

Plants have evolved an intricate molecular response to survive and flourish in their sessile state. Our understanding of these mechanics has come a long way, particularly by using the model plant *Arabidopsis thaliana* [1]. Not only does plant biology enrich the depth of our understanding of principal life processes, but also it is essential for the advancements in the fields of agriculture, industry and biomedical research. Due to its implications for plant performance in general, understanding regulatory circuits underlying root growth and function represents one of the key research areas tackled by the plant sciences, with far-reaching socio-economic implications.

The primary root apical meristem is composed of the meristematic, elongation and differentiation zones, with hormonal cross-talk driving growth and differentiation [2]. The meristematic zone contains the stem cell niche surrounding the quiescent center (QC), which gives rise to new cells that proliferate to the elongation zone and eventually attain their final morphological attributes in the differentiation zone [3]. These portions of the root meristem are defined by morphological features, with the onset of anisotropic cell expansion marking the transition from the meristematic zone into the elongation zone, whilst formation of root hairs is a feature of the differentiation zone [4,5]. Root system architecture is defined by a combination of environmental factors, such as nutrient and water availability as well as the microbiome in the rhizosphere, and depends on intrinsic pathways, made up of a plethora of signaling events that act on expression and activity of regulators of morphogenesis and adaptation [6]. For a long time, auxin [indole-3-acetic acid (IAA) representing its predominant form] was hailed as the central regulator of root development, although it is now clear that root system architecture is governed by an array of hormonal signals, such as cytokinin, gibberellin, ethylene, abscisic acid, brassinosteroids, jasmonates and strigolactone [7,8,9,10,11]. Sensing of and responses to these diverse signals employs quite different pathways, producing a melange of synergistic and antagonistic effects that jointly define root growth. Deciphering the interplay of these various regulators and how they impact on the growth parameters of cells, still encounters major challenges. In this context, the study of directional root growth has proven as extremely valuable tool, as it combines easily accessible plant growth responses with a wealth of resources and techniques established in the *Arabidopsis* model. This has been exploited extensively, resulting in highly valuable insights into pathways and mechanisms driving cellular responses to various environmental inputs.

## 2. Root Gravitropism

Upon germination, roots anchor themselves to grow downwards, thereby facilitating access to water and nutrients [12]. Such directional root growth requires sensing of and responding to gravity signals to ensure-with few exceptions-that roots grow along the gravity vector, referred to as positive gravitropism [13]. Root gravitropism can be divided into distinct steps: gravity sensing, signal transmission and growth response [14]. Perception of variations in the gravity vector primarily involves the root cap, while sites of signal transmission and responses extend into the elongation zone, marking spatial separations in the root for each step [15].

Gravity is perceived by statocytes, which are gravity sensing cells, located in the columella root cap [16]. Consistently, disrupting columella root cap cells using laser ablation causes strong inhibition of the gravitropic response [17]. These statocytes contain starch-filled amyloplasts also known as statoliths. According to the starch-statolith hypothesis, sedimentation of these specialized organelles triggers a signal transduction response, extending further up the root [16,18]. Consequently, tempering with the functionality of statoliths either genetically or physically has been found to interfere with gravitropism [16,19,20,21]. In an alternate protoplast pressure hypothesis, it was suggested that it rather is the pressure of the cytoplasm on the membrane that triggers gravity-induced signaling events [22,23]. As a consequence of such signaling, differential cell elongation predominantly in the root meristem elongation zone, results in a reorientation of root growth to realign with the direction of the gravity vector. However, how does a physical signal, like statolith sedimentation, get translated into a growth determinant? Some theories put forward a critical role for contacts between statoliths and membrane-bound receptors, or an activation of mechanosensitive ion channels [24,25]. The role of actin in the gravitropic sensing response has been studied for many years and was proposed to influence the velocity of statolith sedimentation [26]. From that, it appears that the dynamics in the intracellular statolith distribution, next to statolith-plasma membrane contacts, contributes to initial gravity signaling [27]. Furthermore, and to complicate matters, the gravitropic response in its entirety is not due only to signal perception in the root cap, as it has been shown that events in the elongation zone play a role as well [28,29]. In fact, it appears that different sites and modes of gravity perception have been invented in the plant kingdom, with ‘fast’ gravitropism via root cap statolith signaling introduced only very recently [29,30,31,32]. This diversity is also reflected in the evolution of a repertoire of regulatory switches that jointly coordinate root responses towards gravity.

## 3. Auxin

Sensing of the gravity stimulus ultimately triggers a signaling network orchestrated by the phytohormone auxin, which is key to the coordination of directional root growth in response to gravity [33]. Canonical transcriptional auxin signaling depends on auxin bringing together F-box TRANSPORT INHIBITOR RESPONSE1/AUXIN SIGNALLING F-BOX (TIR1/AFB) and members of the AUXIN/INDOLE-3-ACETIC ACID (Aux/IAA) transcriptional repressor family [34,35]. The F-box proteins are part of a heteromeric SCF-type protein complex that functions as E3 ubiquitin ligase, transferring activated ubiquitin onto Aux/IAA, transcriptional repressors. In this process, both TIR1/AFB and Aux/IAA proteins act as auxin co-receptors, ultimately causing Aux/IAA proteins to be degraded via the proteasome pathway [33,35]. This in turn, releases repression of Auxin Response Element-(ARE)-containing promoters, simply put, as a consequence of derepressed AUXIN RESPONSE FACTOR (ARF) transcription factors no longer dimerizing with Aux/IAAs [36]. Alternate auxin sensing mechanisms have been identified, which involves activity of plasma membrane-resident TRANSMEMBRANE KINASE (TMK) proteins [37,38,39] as well as hormone sensing via the ARF3/ETTIN transcriptional regulator [40,41]. Modes of auxin perception via these pathways are still only partially resolved and matter of discussion [42,43], but downstream signaling events have been characterized to some extent. Specifically, auxin sensing via TMKs affects cytoskeleton remodeling as well as transcriptional reprogramming in context of auxin-controlled cellular differentiation [37,44].

The time it takes for transcriptional changes to take place is not sufficiently rapid for some of the auxin effects described. For example, applying exogenous auxin causes the root to respond within seconds, with a calcium flux and quantifiable apoplastic alkalinization resulting in plasma membrane depolarization [45]. It has been shown that mutants lacking selected TIR1/AFB receptor proteins still continue to have ion flux changes in response to the hormone [35,45,46], highlighting alternate modes of auxin perception and signaling. Furthermore, by using live imaging and microfluidics, it could be demonstrated that auxin effects on root growth are extremely rapid, and that the root shows a highly sensitive readout response to the amount of IAA present in the growth medium [47]. These root growth responses have been linked to non-canonical TIR1/AFB signaling, independently of transcriptional reprogramming [39,47]. Taking all that is known, it is becoming increasingly clear that auxin signaling and transduction involves multifaceted mechanistic arrays that jointly affect cellular growth responses in gravistimulated roots.

## 4. Defining Auxin Signals for a Defined Root Growth Response

In the root meristem, auxin accumulates in proximity of the quiescent center (QC), while hormone concentrations decrease in more distal portions [48]. It is important to note that the auxin concentration does not just change gradually, but rather shows cell-type specific variations, crucial for defined growth responses, such as differential cell elongation in bending roots [49]. Specifically, the epidermal, cortical, and endodermal cell layers show fluctuations in auxin concentration, which have been associated with the hormone’s transit from the root tip towards the elongation zone [49]. Transcriptomics analysis further highlights how cell-specific the auxin dosage responses are, revealing a correlation in responses along the longitudinal axis of the root [50]. Crosstalk with additional growth regulators complicates matters, exemplified by local auxin maxima being counteracted by cytokinin, to transiently antagonize auxin responses [51]. In addition to this, and in combination with further inputs, crosstalk of auxin signaling with genetic determinants of root morphogenesis such as *PLETHORA* (*PLT*) transcription factors affects fundamental processes in the control of developmental transitions in root meristems [52,53]. All things considered, it appears that variations in auxin homeostasis *per se* provide only limited positional information, but rather function in conjunction with additional hormonal input and associated genetic pathways to jointly define root morphogenesis.

The chemiosmotic hypothesis details the mechanism of how auxin moves from cell to cell via specific transport, which also provides directionality to such transport [54,55]. This, in combination with the Cholodny-Went hypothesis, first postulating that asymmetric auxin redistribution in tissues causes differential growth [56], provided the mechanistic foundation for auxin’s function in gravitropism. Indeed, experimental evidence has been put forward to show that polar auxin transport (PAT) generates intercellular auxin gradients guiding root growth and bending [57]. Canonical PIN-FORMED (PIN) cellular auxin efflux proteins, are intimately involved in these processes, facilitating directional cell-to-cell auxin transport as a result of their asymmetric polar localization at the plasma membrane [58,59]. Auxin is distributed equally on all sides of the root meristem when the root is positioned vertically downwards. However, as the root changes direction of growth in response to gravistimulation, so does the auxin gradient, creating more pronounced outputs at the concave, lower side of a bending root. This was shown experimentally by tracking radiolabeled IAA [60], as well as by determining expression of auxin-responsive reporter genes in gravistimulated roots over time [61,62,63,64]. Differential cell elongation is the ultimate consequence of the lateral auxin gradients in root meristems, causing the root to bend in accordance with the gravity vector. In mechanistic terms, local variations in cell wall architecture allow for reversible adjustments in cell expansion. This to some extent appears to be accomplished by auxin-induced membrane depolarization at the gravistimulated root’s lower side, with IAA^-^/proton symport and auxin-induced Ca^++^ uptake by CYCLIC NUCLEOTIDE-GATED CHANNEL 14 (CNGC14) contributing to signaling events, resulting in localized increases in the apoplastic pH [46,65,66,67].

A transient perturbation of the auxin distribution in the root tip is key to auxin-mediated adjustments in directional root growth. This is brought about by altering the localization of components of the auxin transport machinery in gravistimulated columella root cap cells. Specifically, through the accumulation of otherwise apolarly localized PIN3 and PIN7 auxin efflux facilitators to the lower plasma membrane domains of these gravistimulated cells [68,69] (Figure 1). This causes auxin to relocate to the bottom side of the root tip, acting as a trigger for downstream events that guide kinetics of root bending. In mechanistic terms, it still remains unclear as to how this polar PIN relocation is brought about in response to gravity. Advances in the understanding of mechanisms that connect gravity sensing to PIN relocation have been made though. The process involves PIN transcytosis in an ADP ribosylation factor guanine nucleotide exchange factor-(ARF GEF)-dependent manner, together with variations in the PIN phosphorylation status, which influences kinetics and quantities of PIN accumulation at polar plasma membrane domains [70,71,72] (see below, for further details).

Furthermore, ALTERED RESPONSE TO GRAVITY1 (ARG1) and ARG1-LIKE 2 (ARL2), peripheral membrane proteins have been described as essential for the initial perturbation of auxin distribution. *Arg1* and *arl2* mutant lines show deficiencies in root gravitropism, together with a decelerated kinetics in gravity-induced PIN relocation in root cap cells [73,74,75] (Figure 1). Both proteins contain conserved J-domains and were found to interact with chaperone protein HSC70, suggestive of roles in vesicular trafficking as implicated for HSC70 and further J-domain proteins [76]. It is currently not known if such activity could participate in polar PIN transcytosis, which is also true for potential crosstalk between ARG1 and the actin cytoskeleton [76]. Thus, whilst it is tempting to speculate about roles for ARG1/ARL2 in connecting gravity signals to the cytoskeleton and associated adjustments in vesicular protein sorting, experimental evidence is still scarce.

Only recently, members of the enigmatic *LAZY1* (*LZY1*) gene family were found to play a role in polar PIN sorting, creating a connection between statolith sedimentation and auxin distribution via their function in root cap cells [77,78] (Figure 1). Specifically, aberrations in Gravitropic Set Point Angle (GSA) control of lateral roots found in *lzy* loss-of-function mutant combinations, coincided with deficiencies in PIN3 expression/distribution control in lateral root meristems [77]. However, how are LAZY and PIN3 functions interconnected? Members of the RCC1-(Regulator of Chromosome Condensation)-LIKE DOMAIN (RLD) protein family might provide such a link, as their C-terminal domain was found to mediate interaction with the C-terminal CCL-domain of LAZY [79]. This interaction could recruit RLD proteins to the plasma membrane in a LAZY-dependent manner, which, upon gravistimulation, results in RLD/LAZY accumulation at the bottom side of root cap cells of lateral roots. RLD/LAZY relocation occurs in close succession to the sedimentation of statoliths, perhaps as a result of sedimentation-induced signaling. RLD/LAZY relocation in turn is followed by adjustments in PIN protein distribution in lateral root tip cells, which is likely responsible for coordinating auxin distribution upon GSA establishment [79].

Different molecular determinants appear to impact on the efficacy of PIN relocation in gravistimulated roots, like the potential involvement of ARG1 and ARL2 in cytoskeleton-dependent sorting of PINs. On the other hand, LAZY and RLD proteins might represent constituents of a protein complex defining plasma membrane domains for asymmetric recruitment of PINs. In that respect, these proteins might function analogous to BREAKING OF ASYMMETRY IN THE STOMATAL LINEAGE/BREVIS RADIX (BASL/BRX) polarity modules, required for coordination of asymmetric cell divisions and correct differentiation of stomatal cell lineages [80]. Future research can be expected to uncover crosstalk between, and modes of action by which ARG1/ARL2 and LAZY/RLD contribute to the polar targeting of PINs.

## 5. Transmission and Resetting of the Gravity Stimulus

PIN-mediated auxin redistribution to the bottom side of the root tip, initiates a cascade of events that control the timing and the magnitude of the root’s response to gravity. Following stimulus perception and localized auxin rerouting, a root will start bending as a result of auxin asymmetry in the root meristem. However, before completion of root reorientation a tipping point mechanism kicks in, resetting the auxin gradient long before the direction of root growth has realigned to the direction of the gravity vector [62]. Such resetting appears equally important as the initial establishment of an auxin gradient in the root tip, preventing uncontrolled overbending of gravity-responding roots. Sensing and responding to this tipping point involves adjustments in sorting and/or activity of the auxin transport machinery in the root cap. Apart from that, two additional proteins define root bending kinetics via transmission of instructive auxin signals from the root tip into the elongation zone: the auxin cellular uptake facilitator AUXIN RESISTANT 1 (AUX1) and PIN2-mediated cellular auxin efflux. Expression of both genes in the outer cell files of the root meristem is indispensable for gravitropic root bending, indicated by severe agravitropic root growth phenotypes of *aux1/pin2* loss-of-function mutants, more prominent than those of a loss of PINs that act in auxin distribution in the columella root cap [63,81,82,83].

PIN2 and AUX1 function in a cell-to-cell auxin transport mode from the root tip into the elongation zone via successive cellular uptake and efflux steps. It is thus the interplay between these activities that influences auxin flux levels versus intracellular auxin retention time, both defining differential cell expansion and hence root bending. Regulation of expression, subcellular distribution and activity of these transport proteins, therefore can be considered central to mediating auxin effects on gravitropic root bending.

Surprisingly little is still known about the regulation of AUX1 in the control of root gravitropism [84]. Early reports demonstrated that expression of AUX1 in root epidermis and lateral root cap cells as well as its controlled exocytotic sorting to polar domains at the plasma membrane is essential for proper root bending [81,85,86]. Furthermore, variations in *AUX1* transcription have been implicated in the control of shootward auxin transport, exemplified by cytokinin-induced adjustments in *AUX1* abundance in epidermis and lateral root cap cells [87,88]. Resulting modifications in auxin transport were suggested to spatially modulate auxin steady-state levels to affect cell elongation and to contribute to differential growth in gravity-responding roots [88].

PIN2 on the other hand, has meanwhile been shown to be subject to a range of regulatory inputs, adjusting its expression and subcellular distribution [89,90,91,92,93,94,95,96,97,98]. The protein exerts a strictly polar plasma membrane localization at the apical domain of lateral root cap cells (LRC), as well as epidermis and cortex cells distal to the root tip, defining the direction of shootward auxin flux that is essential for the control of differential cell expansion in gravity-responding roots [82,99]. Notably, in cortex cells proximal to the root stem cells PIN2 exhibits a basal plasma membrane localization, which could promote rootward auxin flow. A mechanism of auxin reflux has been put forward to explain this distinctiveness, in which auxin is circulated between cell files in the root meristem, to enhance localized auxin responses uncoupled from external auxin input [100]. Apart from polarity control, PIN2 is also target of a variety of signaling pathways essential for translation of environmental signals into spatiotemporal adjustments in protein abundance, distribution and turnover. Elucidation of such pathways, produced substantial insights into modes of action by which PIN2 modulates auxin flow and defines root growth in response to a range of stimuli [96,97] (Figure 1).

## 6. PIN Expression Control at the Interface of Gravitational Stimuli and Intrinsic Cues

Unlike PIN3, PIN4 and PIN7, which are instrumental for establishing perturbations in the distribution of auxin in the very tip of the root, PIN2 acts in the transmission of such hormonal imbalances. In particular, its predominantly apical localization in cells of the root meristem is central for polar auxin delivery into the root elongation zone [99]. PIN2 abundance at the plasma membrane is subject to an interplay between exocytotic sorting, re-internalization via clathrin-mediated endocytosis (CME), followed either by recycling to the plasma membrane or by ubiquitylation-dependent vacuolar degradation [101,102,103,104,105]. Resulting variations in PIN2 plasma membrane homeostasis participate in the regulation of directional auxin flow, specifically in roots responding to environmental and metabolic stimuli (Figure 1). High salinity in the rhizosphere represents a stimulus that promotes such differential sorting of PIN2. Specifically, root halotropism, i.e., adaptation of root growth to avoid saline environments- requires directional root growth that coincided with a lateral PIN2 gradient in the root meristems [106]. Gravistimulation is another example where roots respond by forming a lateral PIN2 gradient, with more protein at the plasma membrane of cells at the lower side of such roots (Figure 1). This gradient is transient and no longer detectable during later stages of gravitropic root bending [107,108]. Furthermore, and consistent with a role for PIN2 sorting in adaptive root growth responses, disturbance of PIN2 endocytic sorting by *cis*-acting mutations affecting either PIN2 ubiquitylation or crosstalk with AP-2 sorting receptors, impacts on the protein’s ability to rescue *pin2* loss-of-function mutants [102,104].

In general, it seems that a range of stimuli contributes either to establishment or resolution of lateral PIN2 gradients, aiding fine tuning of auxin flux rates at upper and lower sides of gravity-responding roots. For example, elevated GA levels in cells at the root’s lower margin are postulated to cause PIN2 stabilization, indicated by a transient maximum of GA signaling at the lower side of gravistimulated primary root meristems that coincides with a local increase in PIN2 protein abundance [109]. Consistent with such GA effects, pharmacological or genetic interference with GA responses results in altered root growth, including deficiencies in gravitropism, potentially caused by deregulated sorting and/or turnover of PINs [109,110]. Models propose that GA and additional growth regulators/stimuli reinforce differential auxin transport and root bending via promoting the formation of lateral PIN2 gradients [109]. Assessment of crosstalk between PIN2 and brassinosteroid responses however has put this concept into question. By analogy to GA effects, the plant steroid hormone has been found to cause stabilization of PIN2 by antagonizing its endocytic sorting to the lytic vacuole [111]. In addition, hormone treatment was found to interfere with formation of a PIN2 gradient in gravistimulated roots, likely as a result of protein stabilization. Nevertheless, brassinosteroid-induced PIN2 stabilization did not interfere with formation of an auxin signaling gradient, and root downward bending in response to gravistimulation. In fact, under these growth conditions, roots even exhibited exaggerated gravity responsiveness, frequently resulting in root overbending. Lateral PIN2 gradients thus appear dispensable for early phases of the root’s gravitropic response but could rather function in its termination, preventing roots from overbending. Modelling the role of PIN2 steady-state levels in the control of auxin flux, supported this hypothesis [111]. In these simulations, it turned out that asymmetry in PIN distribution in columella root cap cells is sufficient for transmission of an auxin gradient into the root elongation zone. Contrastingly, a lateral PIN2 gradient in root meristems was found to dampen the steepness of this auxin gradient by increasing the intracellular auxin retention due to diminished PIN2 levels at the root’s upper side (Figure 1). Somehow counterintuitive, PIN2 gradient formation thus might antagonize excess buildup of an auxin concentration gradient, which could contribute to a resetting of the differential auxin signaling upon completion of gravitropic root bending.

All these findings reflect only small details of a presumably highly complex network of various synergistic and antagonistic inputs, acting on PIN2 in gravistimulated roots. The responses of PIN2 to auxin itself represent a good example for such multifaceted inputs. Originally auxin was discovered as a signal promoting degradation of PIN2 by inducing its ubiquitylation [104,112], with further studies implying a role for auxin in stabilizing PIN2 at the plasma membrane [113]. Apart from that, a quantification of PIN2 signals in gravistimulated roots indicated that both super- as well as suboptimal auxin concentrations induce PIN2 degradation in a defined spatiotemporal context [114]. From that it appears that a diverse input of auxin-stimulated signaling events contribute to initiation and resetting of gravitational root bending via control of PIN2.

PIN2 auxin responsiveness appears to be regulated by canonical SCF^TIR1/AFB^ signaling as well as by pathways involving plasma membrane-localized TMKs [112,114,115]. TMK-PIN2 crosstalk is highlighted in a recent study, describing the characterization of MEMBRANE ASSOCIATED KINASE REGULATOR2 (MAKR2), a plasma membrane-associated protein, with an apparent role in the control of TMK1 kinase activity [116]. Loss of *MAKR2* causes stabilization of PIN2 at the lower side of gravistimulated roots, possibly due to abolished crosstalk between TMK1 and RHO OF PLANT 6 (ROP6), and coinciding with exaggerated root bending in *makr2* silencer lines. Elevated auxin steady-state levels on the other hand cause MAKR2 dissociation from the plasma membrane into the cytoplasm in a TMK-dependent manner. This relieves TMK function in the control of ROP6 and hence PIN2 sorting, a process seemingly essential for coordinating pace and dynamics of gravitropic root bending [116,117].

The auxin example adequately highlights the diversity of regulatory switches by which a single growth regulator could affect PIN function at transcriptional and various post-transcriptional layers of regulation. Unquestionably, a similar complexity can be expected for the input of additional growth regulators and stimuli in the control of PINs. From what we have learned so far about the role of PIN2 in root gravitropism, it seems that, for the most part, all these findings represent only isolated bits and pieces. Putting these into context, by studying the combinatorial input of the diverse PIN2 effectors, and by considering effects of determinants of auxin distribution and root gravitropism that act in conjunction with PIN2, remains a challenge for future research.

## 7. PIN Membrane Sorting and Polarity Control

Adjustments in PIN activity and abundance are efficient means to control auxin flux rates. However, modifications in the directionality of auxin distribution, require alterations in PIN polar distribution at the plasma membrane as implicated throughout plant morphogenesis, as well as in comparably swift adjustments in directional organ growth, such as gravitropism [68,118]. Fundamental sorting mechanisms have been connected to the regulation of PIN polarity in root meristem cells. These involve sorting decisions controlled by ARFs, which mediate cargo sorting to polar plasma membrane domains and recycling via elements of the TGN. Detailed analyses of an ARF GEF termed GNOM provided mechanistic insights into these particular processes. GNOM influences vesicle formation at cellular membranes by controlling the activity of selected ARFs [119]. It belongs to the Brefeldin A-(BFA)-sensitive ARF GEFs and impacts primarily plasma membrane targeting and recycling of cargo sorted to the basal domain. A prominent cargo is PIN1, which is expressed in the root stele and facilitates rootward auxin transport [120]. Interference with GNOM function, either in *gnom* loss-of-function mutants or by BFA treatment, resulted in the internalization of otherwise basally localized PIN1, associated with severe defects in auxin-controlled plant morphogenesis [120]. Apically localized PIN2 on the other hand, exhibited comparably limited responsiveness to reduced GNOM functionality, underlining roles for GNOM, preferentially in the basal sorting of PINs, but also raising questions about corresponding pathways that might function in apical PIN sorting pathways [70].

PIN protein phosphorylation, at conserved sites in the central hydrophilic loop domain, by members of a family of serine/threonine kinases (Figure 2), appears essential for apical-basal sorting decisions [121]. Activity of AGCVIII-type kinase PINOID (PID) was originally found to regulate localization of PIN, with loss of *PID* and its close relates *WAG1* and *WAG2* causing basalization of PINs whilst the opposite effect can be observed in *PID* overexpression lines, promoting PIN apicalization [122,123]. PID/WAG1/WAG2 phosphorylate PINs at serines found in three loop-resident TPRXS(N/S) motifs (Figure 2A), antagonized by the phosphatase activity of PP2A/PP6 proteins [121,124]. Phospho-dead *pin1* alleles mutated in these sites cause developmental defects as seen in the *pid* mutant, presumably due to mislocalized PIN1 in developing inflorescence axes [125]. Similarly, corresponding *pin2* phospho-dead alleles no longer rescued a *pin2* loss-of-function allele, and showed basal instead of apical subcellular localization in root meristem cells [122], underlining PID/WAG functions in PIN polarity control. Additional protein kinase activities have been implicated in the control of PIN proteins, affecting rate and directionality of auxin flow [126,127,128]. CAMEL (CANALIZATION-RELATED AUXIN-REGULATED MALECTIN-TYPE RLK) and CANAR (CANALIZATION-RELATED RECEPTOR-LIKE KINASE) are so far unique among these kinases, as they encode plasma membrane-localized leucine-rich repeat receptor-like kinases (RLK) [129]. CAMEL was found to catalyze PIN1 phosphorylation at distinct serines and threonines in the central hydrophilic loop domain, with CANAR possibly representing a pseudokinase functioning antagonistically to CAMEL. Such phosphorylation governs PIN1 polar localization in response to variations in auxin homeostasis, a process implicated in the polarization of individual cells or entire cell files, upon vasculature differentiation [130]. CAMEL affects subcellular trafficking and auxin-controlled PIN polarity acquisition in root meristem cells as well, representing another mode of regulation by which auxin could impact on its own distribution by manipulating its transport machinery [129,131]. Identification of extracellular ligands recognized by CAMEL and/or CANAR, will hopefully shed light on processes upstream of the polar PIN sorting machinery.

Differing affinities for phosphorylated vs. non-phosphorylated PIN proteins could modulate their routing into a basal GNOM-dependent pathway, with PINs phosphorylated by PID/WAG representing poor substrates for such sorting, resulting in their predominantly apical localization [70]. Characterization of the AGCVIII kinase subclade of D6 PROTEIN KINASEs (D6PK), challenged this straightforward mode for PIN polarity acquisition control. These kinases exhibit PIN phospho-site specificity overlapping with that of PID/WAGs, although D6PK-mediated phosphorylation appears dispensable for PIN polarity control [132,133]. Instead, phosphorylation by either PID/WAGs or D6PKs was found to induce PIN auxin transport activity, arguing for functions of these kinases different from those in PIN polarity control. Detection of phosphorylated PIN1 at the basal domain of root meristem cells, by antibodies designed to specifically recognize PIN1 phosphorylated by PID/D6PK, represents another inconsistency, as such phosphorylation is assumed to favor apical PIN localization [134]. It appears that next to PID/WAG- and PP2A/PP6-mediated phosphorylation control, additional molecular determinants are needed to define PIN polarity and hence directionality of auxin transport.

## 8. Plasma Membrane and Intrinsic Determinants in PIN Polarity Control

Phosphoinositides (PIP) are membrane phospholipid components, which, although they make up less than 1% of total membrane lipid composition, define membrane identity, as they account for many critical processes, such as protein recruitment, cytoskeletal dynamics, enzyme activity, and membrane trafficking regulation [135,136]. Certain PIPs have been associated with different membranous structures in plant cells: PI3P and PI(3,5)P_2_ in late endosomes, PI4P at plasma membrane and early endosomes, and PI(4,5)P_2_ at the plasma membrane, with PIPs at the plasma membrane showing a moderately polar enrichment at the apical and basal domains [137,138,139] (Figure 2A,B). A similar polar distribution has also been described for members of the PHOSPHATIDYLINOSITOL-4-PHOSPHATE 5-KINASE (PIP5K) protein family, specifically PIP5K1 and PIP5K2, arguing for their function in localized PI(4,5)P_2_ synthesis at these plasma membrane domains [138,139] (Figure 2A,B). Remarkably, aberrations in the polar distribution of PIN proteins have been described [53,139]. Disturbances in PIN2 polarity in *pip5k1 pip5k2* could relate to deficiencies in CME, as defects in Clathrin Coated Vesicle (CCV) formation were described in *pip5k1 pip5k2*. [138,139,140,141]. Modes of PIN2 polarity acquisition, with super-polar PIN2 delivery to apical plasma membrane domains, followed by its lateral diffusion and highly localized CME at the margins of polar plasma membrane domains [102], thus could require PIP5K activity in defining sites of PIN2 internalization (Figure 2A). PIN1 polar targeting to the basal plasma membrane domain of root stele protophloem cells, likely also involves crosstalk with PIPs (Figure 2B). Here, it appears that AGCVIII-type PROTEIN KINASE ASSOCIATED WITH BRX (PAX) a positive regulator of PIN1 [127], together with BREVIS RADIX (BRX), recruits PIP5K to the basal plasma membrane domain (Figure 2B). This is suggested to cause variations in PI(4,5)P_2_ abundance as a consequence of locally differing PIP5K activity. As a result, spatially differing rates of CME would remove PIN1 at distinct plasma membrane microdomains [142]. Protein distribution consistent with such a setting has been described for the basal pole of root meristem protophloem cells. Here, PAX, BRX and PIP5K1 are concentrated in a central ‘muffin’-shaped domain with limited amounts of PIN1. These ‘muffins’ are surrounded by a ‘donut’-shaped structure enriched for PIN1, presumably representing sites of less efficient endocytosis [142]. It is not entirely resolved as to how such PIN localization might influence auxin distribution, but quite remarkably the ‘donut’ pattern of PIN1 distribution has so far only been described for root meristem protophloem cells, perhaps contributing to the differentiation of these highly specialized cell files [142].

Next to PIPs, PIN polarity control at the plasma membrane is related to additional *trans*-acting determinants, involving secondary messengers, such as cytosolic calcium [126,143]. Cell wall integrity also plays an essential role in these processes. A forward genetic screen aimed at *REGULATORS OF PIN POLARITY* (*REPP*) genes which, when mutated, cause a basal to apical switch of ectopically expressed PIN1 in epidermis cells, led to the identification of *repp3,* deficient in *CELLULOSE SYNTHASE A3* [144]. This was further supported by the circumstance that enzymatically induced cell wall degradation, caused loss of PIN polarization. Strikingly, limited plasmolysis of root meristem cells revealed association of PIN reporter protein signals with Hechtian strands, signifying cytoplasmic connections between PIN proteins and some unknown cell wall-resident determinant, possibly affecting protein conformation and diffusion within the plasma membrane continuum [144,145].

It is important to note that some PINs, most prominently PIN2, show a strong tendency to accumulate in cluster-like structures at the plasma membrane [102] (Figure 2C). Such “clustered” proteins appear to exhibit only a moderate membrane mobility, which in case of PIN2 might contribute to maintaining its highly polar localization at the apical domain [102,146]. Furthermore, polar proteins show higher tendency of clustering when compared to non-polar plasma membrane proteins. Thus, variations in clustering could affect protein diffusion as a means to regulate their polar distribution [147]. However, how is such PIN clustering and intramembrane mobility controlled? Sterol content of membranes, which is long known to impact on PIN polarity [102,148,149], has been suggested to participate [147] (Figure 2C). Sterols are a key factor in determining lipid raft/nanodomain organization in membranes by affecting formation of liquid-ordered (*L*_o_) phase areas versus less tightly packed liquid-disordered (*L*_d_) phase areas [150]. Coordinated structuring of *L*_o_ domains are essential for the ordered formation of nanodomains, with membrane-anchored proteins, such as the plant-specific remorins (REM), participating in this process (Figure 2C) [151]. Recently, *Arabidopsis* REMs were found to promote nanodomain organization as an integral element of salicylic acid-(SA)-controlled plant defense responses [152]. SA also affects endocytic sorting of PIN2, a response suggested to result from inhibitory SA effects on CME [153]. A closer look on SA-PIN2 crosstalk produced some amazing insights, as it turned out that SA impacts on PIN2, potentially via its role in nanodomain organization [154]. In this study, it was demonstrated that SA promotes the clustering of PIN2 at the plasma membrane in a remorin-dependent manner, counteracting PIN2 lateral diffusion and consequently its CME (Figure 2C). The resulting interference with PIN2 distribution coincided with deficiencies in auxin distribution and root growth in response to SA, highlighting pathways by which variations in membrane lipid organization would impact on PIN function via regulation of nanodomain organization. Intriguingly, REM recruitment to plasma membrane domains in SA-controlled defense responses was connected to the activity of 14-3-3 proteins [152]. A related role has been ascribed to 14-3-3 epsilon group members in PIN2 polarity establishment [155] (Figure 2C). It remains to be determined whether REMs and 14-3-3 proteins function in overlapping pathways that control PIN clustering at the plasma membrane.

Microtubule organization and cell wall integrity might also participate in PIN clustering control; this relationship however remains to be elaborated in more detail [147]. This is also the case when it comes to elucidating PIN-PIP crosstalk in control of clustering. Mutant analysis and pharmacological interference revealed an immanent impact of PIP signaling and distribution on the clustering of PIN2 at the plasma membrane. In particular, plasma membrane-resident PIP5K as well as Phospholipase C (PLC) activity appear to be involved, promoting PIN2 clustering and consequently influencing its polar localization [147]. Stimulus-induced variations in PIP5K or PLC expression or activity thus appear to contribute to PIN regulation, representing yet another mode of dynamic regulation of PIN distribution in, for example, gravistimulated roots. Nevertheless, enhancement of PIN polar plasma membrane localization via PIP-induced protein clustering is in marked contrast to the above-described function of selected PIPs in promoting early steps of CME of plasma membrane proteins [139]. Perhaps, this alternate role for PIPs in the regulation of PIN clustering is related to another unanticipated observation, as the PIN2 hydrophilic loop was found to associate with a whole array of PIPs in vitro [147]. This type of interaction is reminiscent of polar targeting of a subset of peripheral plasma membrane proteins, which was demonstrated to depend on interaction between proteinogenic basic amino acid stretches and membrane-resident PIPs [156]. It remains to be determined if similar modes of interaction participate in the polar targeting and clustering of PIN intrinsic membrane proteins.

Structural insights into PIN conformation either in the mobile, dispersed or less mobile, clustered fraction are urgently needed, in order to understand function of and interplay between these differing states. Remarkably, there is strong evidence for hetero- and homo-oligomerization of different plasma membrane-resident *Arabidopsis* PINs, with apparent consequences for configuration and function. A recent study, employing native protein separation techniques, demonstrated PIN oligomer formation, stabilized by inhibitors of directional auxin transport, namely endogenous flavonols and the synthetic compound 1-*N*-naphthylphthalamic acid (NPA). IAA, on the other hand, seems to antagonize these NPA effects, hinting at mechanisms, by which substrate-controlled adjustments in PIN quaternary structure act to regulate auxin transport activity [157]. A related study, provided strong evidence for a direct interaction between NPA and PIN proteins, coinciding with adjustments in the overall conformation of native PIN protein complexes, and unraveling long-searched for insights into modes of NPA-induced inhibition of auxin transport [158].

Whilst these two reports represent an important step forward towards elucidating the native PIN configuration and its relationship to auxin transport, we can only guess about intrinsic determinants, participating in the control of PIN oligomerization. Comparison of PIN conformation under reducing vs. non-reducing conditions, indicated disulfide bridge formation between PIN monomers as such a determinant [158] (Figure 2C). This hints at a role for the redox status of microenvironments at the plasma membrane, in affecting PIN-mediated auxin fluxes, which is in line with observations demonstrating a role for reactive oxygen species (ROS) and redox signaling in the regulation of tropic root growth responses [67,159]. Analysis of a mutant version of PIN2, with both of its cysteines mutagenized, revealed only moderate effects on protein function in its ability to rescue a *pin2* null allele [160]. However, the mutant protein exhibited a less homogenous distribution in plasma membrane microdomains and increased intracellular accumulation, overall indicative of an altered protein mobility. Thus, whilst cysteine-dependent effects on protein conformation appear almost dispensable for PIN2 function, altered protein distribution argue for participation of redox signaling events in PIN sorting control [160] (Figure 2C). Again, further experiments, such as analyzing determinants of PIN multimerization, protein sorting kinetics and crosstalk with additional known effectors of PIN activity, should help clarifying so far elusive regulatory mechanisms of PIN-dependent auxin transport processes.

## 9. Outlook

By exploiting root gravitropism as a simple but highly sensitive readout for monitoring variations in auxin fluxes, stimulating novel insights into the function of PIN proteins have been established in recent years. This involves major advances in our understanding of intracellular protein sorting processes, modes of polar protein targeting as well as first insights into the native conformation of PINs and its relationships to activity in polar auxin transport. However, and as always when it comes to outstanding research, emerging new questions posit demanding challenges for designing future experiments. This relates to systemic approaches, trying to integrate and to resolve the interplay between those various pathways and switches that jointly affect auxin fluxes via controlling PINs. Elucidation of the native PIN conformation in further detail represents another important issue, as are attempts aiming at unraveling composition and function of the enigmatic PIN clusters at the plasma membrane. Ultimately, only a PIN structure at a reasonably high resolution will provide valid answers to at least some of these questions.

## Figures and Tables

**Figure 1 ijms-22-02749-f001:**
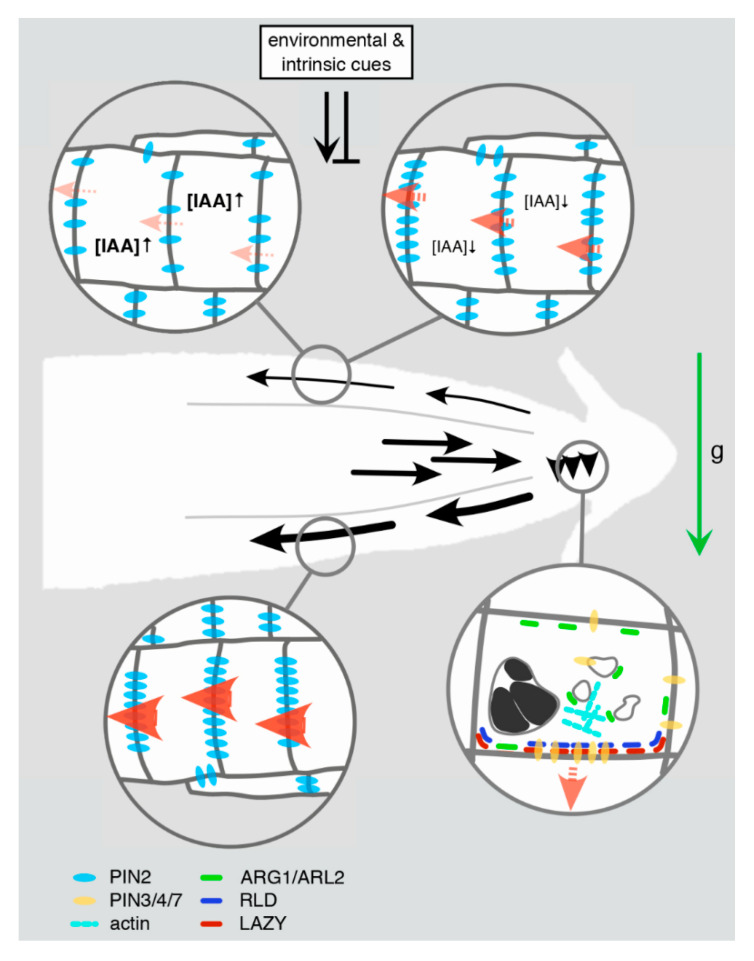
Gravitropic root bending and associated PIN relocation and expression. Upon gravistimulation (“g”, green arrow), a gradient of differential auxin fluxes is transiently established (black arrows and arrowheads), adjusting root growth direction via modulation of cell elongation. Within several minutes after gravistimulation, PIN proteins (dark yellow) in columella root cap statocytes undergo polarization at the plasma membrane at the cellular lower margin. This response appears to be triggered by starch-filled (black structures) statolith movement/sedimentation in accordance with the gravity vector. Molecular players involve ARG1/ARL2 J-domain proteins (green), which might link PIN transcytosis to vesicular transport and/or elements of the actin cytoskeleton (turquoise rods). LAZY (red) and RLD (dark blue) proteins represent another polarity determination module, whose concerted accumulation at the statocytes’ lower margin might precede polar accumulation of PINs. The resulting asymmetry in localized auxin distribution is transmitted into the root elongation zone, with elevated auxin levels transported at the root’s lower side (thick black arrows). A transient increase in the abundance of PIN2 (light blue), predominantly in epidermis cells at the lower side of the root meristem might result from diminished endocytosis and turnover of the protein. This could translate into elevated auxin flux rates (red arrowheads) in these cells, ultimately causing inhibition of cell elongation. At the root meristem’s upper side diminished levels of PIN2 are assumed to reduce auxin flux rates, promoting differential cell elongation at upper and lower and hence bending of the root. Variations in PIN2 abundance in these cells appear to be controlled by a combination of environmental and intrinsic cues, thereby impacting on intracellular auxin steady-state-levels and causing adjustments in the lateral auxin gradient. Such spatiotemporal variations in auxin flux and gradients have been suggested to participate in the resetting of differential auxin transport and gravitational root bending.

**Figure 2 ijms-22-02749-f002:**
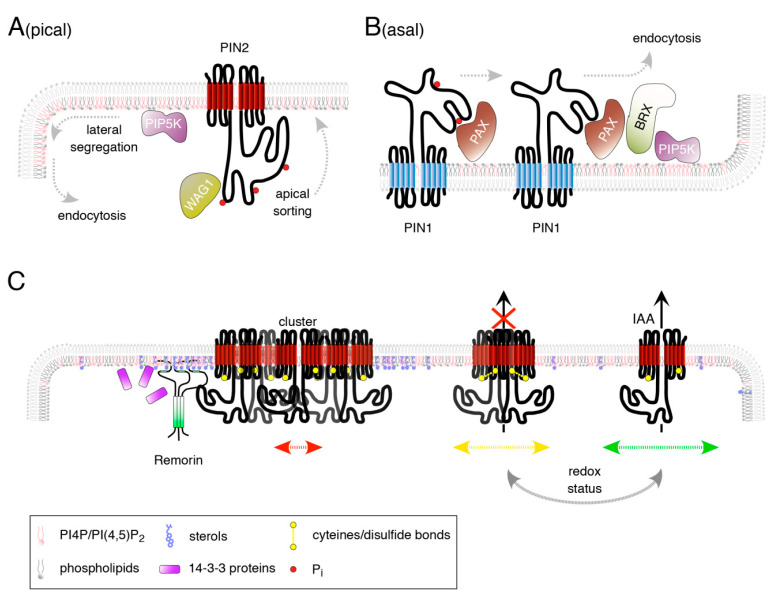
Mechanisms implicated in PIN configuration control and polarity acquisition. (**A**) PIN2 apicalization has been connected to its phosphorylation (red circles) by PID/WAGs at 3 highly conserved loop-resident motifs. Once targeted, PIN2 undergoes lateral diffusion within the plasma membrane continuum. CME taking place predominantly at the apical subdomains of the longitudinal plasma membrane domains appears to be under positive control of increased PI(4,5)P_2_ content in the inner leaflet of the lipid bilayer. Polar enrichment of PIP5K might participate in establishment of local PI(4,5)P_2_ accumulations, aiding CCV maturation. (**B**) PIN1 auxin efflux activity at the basal plasma membrane domain of root meristem protophloem cells is under positive control of PAX-mediated protein phosphorylation (red circles). PAX association with BRX abolishes such activation and furthermore acts as nucleus for localized targeting of PIP5K to central portions of the basal plasma membrane domain. Resulting conversion of PI4P into PI(4,5)P_2_ in turn is believed to promote CME of PIN1. (**C**) Reversible PIN2 recruitment into clustered, immobile protein accumulations (left) might help retain PIN2 in its polar apical plasma membrane domain. PIN2 clustering has been linked to the formation of *L*_o_ phase areas and associated nanodomain formation. These domains are characterized by local enrichment in sterols and PIPs, aiding anchoring of remorin oligomers (green) and further structuring of PIN2-containing nanodomains. 14-3-3 proteins have also been implicated in polarized distribution control of PIN2 and might participate in REM recruitment. Apart from cluster formation, PIN oligomerization has recently been suggested to be under redox control (middle/right), hypothetically involving disulfide bridge formation between highly conserved cysteines found in PIN2 (yellow circles). Variations in the proportion of plasma membrane-associated oligomeric vs. monomeric PINs might influence auxin transport activity and might as well affect intramembrane mobility (indicated by bidirectional green, yellow and red arrows; representing high, intermediate and moderate mobility).

## Data Availability

Not applicable.

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
