# Peer review of "Auxin and Root Gravitropism: Addressing Basic Cellular Processes by Exploiting a Defined Growth Response"

_ijms, 2021, doi:10.3390/ijms22052749_

Round 1

Reviewer 1 Report

The paper reads well, it is clearly organized and updated. The level of synthesis is fine and a critical analysis of available data should be particularly mentioned. It is instructive and stimulates perspective studies. I have only two minor observations.

  1. There are too many abbreviations. Although, these are desglosed at the first use, their following up is not easy. Understanding and reading may be facilitated if the authors  provide the complete list of abbreviations at the beginning.
  2. Fig. 1. It is not obvious without main text that the figure depicts a gravitropic root bending. Please, indicate it in the figure legend.  Also, albeit it may be self-explicable, please indicate that g and green arrow show the direction of gravity stimulus.    

Author Response

Point-by-point response:

Reviewer 1:

The paper reads well, it is clearly organized and updated. The level of synthesis is fine and a critical analysis of available data should be particularly mentioned. It is instructive and stimulates perspective studies. I have only two minor observations.

  1. There are too many abbreviations. Although, these are desglosed at the first use, their following up is not easy. Understanding and reading may be facilitated if the authors provide the complete list of abbreviations at the beginning.

Response:

We added a complete list of abbreviations, which currently can be found at the end of the body text (since we are not familiar with the journal's guidelines on where to put this list).

  1. Fig. 1. It is not obvious without main text that the figure depicts a gravitropic root bending. Please, indicate it in the figure legend. Also, albeit it may be self-explicable, please indicate that g and green arrow show the direction of gravity stimulus.

Response:

We have changed the figure legend accordingly.

Reviewer 2 Report

This manuscript entitled “Auxin and root gravitropism: Addressing basic cellular processes by exploiting a defined growth response” by Konstantinova et al. comprises the necessary elements of scientific novelty.

Below are some minor comments.

  • The authors should double-check their language and grammar. Some of the descriptions are not precise, which will need to be rephrased.
  • Comment 1: Line 17-20, Try not to write lengthy sentence and rewrite
  • Line 28-30: rewrite the sentence as it is not connecting the former sentence properly
  • Line 35-37: There are apparent …. Zone; correct and rewrite the sentence
  • Line 45 to 47: Deciphering …challenges; correct the grammatical errors
  • Comment 2: author’s need to establish the importance of this review with more supporting data (in between line 44-45) in the ‘Background’ section.
  • Line 54: This directional …signals, correct it
  • Line 55-58: Gravitropism … Gravitropism, rewrite
  • Line 59-61: sensed? A sentence needs to be Written in detail
  • Line 64-67: lengthy sentence, rewrite
  • Line 78-80: so… well; rephrase
  • Line 95: E3 meant for?
  • Line 98-103: rephrase the sentence
  • Line 105-107: incomplete sentence
  • Line 123: an auxin?
  • Line 164-167: rewrite
  • Line 169 to 220: lacks clarity, try to minimize writing lengthy big sentences
  • Comment 3: in the ‘PIN membrane sorting and polarity control “and
  • ‘Plasma membrane and intrinsic determinants in PIN polarity control ‘section. Minimize the Repetition of information and needs language editing.

Author Response

Reviewer 2:

This manuscript entitled “Auxin and root gravitropism: Addressing basic cellular processes by exploiting a defined growth response” by Konstantinova et al. comprises the necessary elements of scientific novelty.

Below are some minor comments.

The authors should double-check their language and grammar. Some of the descriptions are not precise, which will need to be rephrased.

Response:

Thank you very much indeed for highlighting this issue. We carefully went through the text and did some extensive editing and rephrasing, in order to avoid ambiguousness.

Comment 1: Line 17-20, Try not to write lengthy sentence and rewrite

Response: Has been rewritten in the revised manuscript.

Line 28-30: rewrite the sentence as it is not connecting the former sentence properly

Response: We changed this sentence accordingly.

Line 35-37: There are apparent .... Zone; correct and rewrite the sentence

Response: has been edited.

Line 45 to 47: Deciphering ...challenges; correct the grammatical errors

Comment 2: author’s need to establish the importance of this review with more supporting data (in between line 44- 45) in the ‘Background’ section.

Response: We modified this section to further underline the 'message' of our review.

Line 54: This directional ...signals, correct it

Response: we edited this section.

Line 55-58: Gravitropism ... Gravitropism, rewrite

Response: We edited this section.

Line 59-61: sensed? A sentence needs to be Written in detail

Response: We further specified this information in the revised manuscript.

Line 64-67: lengthy sentence, rewrite Line 78-80: so... well; rephrase

Response: This has been rephrased accordingly

Line 95: E3 meant for?

Response: According to Hershko and colleagues (1983; J Biol Chem. 258:8206-14; 1998; Annu. Rev. Biochem. 67:425–79); 'The first E3 discovered, E3α, was originally defined operationally as a third enzyme component required, in addition to E1 and E2, for the ligation of ubiquitin to some specific proteins.'

Line 98-103: rephrase the sentence Line 105-107: incomplete sentence

Response: We rephrased this sentence.

Line 123: an auxin? Line 164-167: rewrite

Response: We took care of that issue

Line 169 to 220: lacks clarity, try to minimize writing lengthy big sentences

Comment 3: in the ‘PIN membrane sorting and polarity control “and

‘Plasma membrane and intrinsic determinants in PIN polarity control ‘section. Minimize the Repetition of information and needs language editing.

Response: We carefully went through this section of the manuscript as pointed out by the reviewer. We took care of lengthy sections and streamlined these parts of the manuscript.